# Short-Term Associations of Ambient Fine Particulate Matter (PM_2.5_) with All-Cause Hospital Admissions and Total Charges in 12 Japanese Cities

**DOI:** 10.3390/ijerph18084116

**Published:** 2021-04-13

**Authors:** Kohei Hasegawa, Hirokazu Toubou, Teruomi Tsukahara, Tetsuo Nomiyama

**Affiliations:** 1Department of Preventive Medicine and Public Health, School of Medicine, Shinshu University, 3-1-1 Asahi, Matsumoto, Nagano 390-8621, Japan; htoubou@shinshu-u.ac.jp (H.T.); tsukat@shinshu-u.ac.jp (T.T.); nomiyama@shinshu-u.ac.jp (T.N.); 2Department of Occupational Medicine, School of Medicine, Shinshu University, 3-1-1 Asahi, Matsumoto, Nagano 390-8621, Japan

**Keywords:** air pollution, particulate matter, environmental pollutants, short-term exposure, morbidity, hospital admission, health care cost, economic cost, time series analysis, environmental epidemiology

## Abstract

The short-term association between ambient particulate matter ≤2.5 microns in diameter (PM_2.5_) and hospital admissions is not fully understood. Studies of this association with hospital admission costs are also scarce, especially in entire hospitalized populations. We examined the association between ambient PM_2.5_ and all-cause hospital admissions, the corresponding total charges, and the total charges per patient by analyzing the hospital admission data of 2 years from 628 hospitals in 12 cities in Japan. We used generalized additive models with quasi-Poisson regression for hospital admissions and generalized additive models with log-linear regression for total charges and total charges per patient. We first estimated city-specific results and the combined results by random-effect models. A total of 2,017,750 hospital admissions were identified. A 10 µg/m^3^ increase in the 2 day moving average was associated with a 0.56% (95% CI: 0.14–0.99%) increase in all-cause hospital admissions and a 1.17% (95% CI: 0.44–1.90%) increase in total charges, and a 10 µg/m^3^ increase in the prior 2 days was associated with a 0.75% (95% CI: 0.34–1.16%) increase in total charges per patient. Short-term exposure to ambient PM_2.5_ was associated with increased all-cause hospital admissions, total charges, and total charges per patient.

## 1. Introduction

Exposure to ambient air pollution is associated with adverse health effects [1,2]. Among the many pollutants, exposure to ambient particulate matter with an aerodynamic diameter of ≤2.5 μm (PM_2.5_) is a critical risk factor for mortality and morbidity. It was estimated that 4.2 million deaths, approximately 7.6% of total deaths worldwide, are attributable to ambient PM_2.5_ [3].

Several research groups have reported that exposure to ambient PM_2.5_ is associated with the risk of hospital admission, but their studies focused on pulmonary or cardiovascular diseases [4,5,6,7]; more recent studies revealed that ambient PM_2.5_ exposure is associated with a broader range of diseases [8,9]. Studies of the relationship between ambient PM_2.5_ exposure and all-cause hospital admissions are limited [6,10]. In addition, studies of the association between hospital admissions costs and ambient PM_2.5_ exposure are scarce, and the existing studies targeted specific disease groups [11,12,13]. It is necessary to conduct studies that target entire hospitalized populations in order to clarify the economic burden of ambient PM_2.5_.

In Japan, automated ambient PM_2.5_ measurement by standardized methods began around 2012, enabling nationwide studies of the health effects of ambient PM_2.5_ exposure [14,15]. In 2017, the Ministry of Health, Labour and Welfare of Japan (MHLW) started to provide Diagnosis Procedure Combination (DPC) [16,17] survey data in aggregated form. In the survey, MHLW collected simplified medical records of each hospitalized patient from most of the acute-care hospitals in Japan. These data were used to examine the nationwide occurrence of cryptorchidism [18].

By combining these newly available data, we examined the association between short-term exposure to ambient PM_2.5_ and all-cause hospital admissions and the corresponding total charges [19]. We also examined the total charges per patient to investigate whether ambient PM_2.5_ exposure before the hospital admission event was associated with the subsequent total charge from the admission. We conducted the present study in Japan, where the PM_2.5_ level is one of the lowest in Asia [20].

## 2. Materials and Methods

### 2.1. Study Area and Period

This study included the 12 cities in Japan (Appendix A) for which the total population was >1 million as of the 2015 national census [21]. The 12 cities are located throughout the length of Japan. All of the cities have more than one ambient air pollution monitoring station with automated PM_2.5_ measurement. Table 1 provides the population and size of the 12 cities. In all of the cities, the study period was from 1 April 2015 to 31 March 2017.

### 2.2. Health Data

The DPC is a classification method developed in Japan for inpatients in the acute phase [16,17]. The diagnostic procedure combination/per-diem payment system (DPC/PDPS) is a reimbursement system based on the DPC for acute inpatient care. As of 2015, this payment system was adopted in 1580 hospitals in Japan. These hospitals are called DPC hospitals and cover most acute inpatients [17].

To evaluate and update the DPC/PDPS, the MHLW distributed the DPC survey every year among hospitals that had already adopted the DPC/PDPS and those that were preparing to adopt the DPC/PDPS. In the survey, MHLW collected simplified medical records of all of the discharged patients from the hospitals in the calendar year. The collected data include age, sex, place of residence, the primary diagnosis coded in the 10th edition of the International Classification of Disease (ICD-10), the medical procedures performed, and the drugs administered.

We applied to the MHLW and obtained the DPC survey data in the aggregated form. The provided data included the number of hospital admissions and corresponding total charges stratified by patient age (<65 and ≥65), sex, date of admission, and city. To prevent the identification of individual patients, the MHLW applied an anonymization process before they provided the data to us. Specifically, if the number of patients in a cell was <10, the cell’s value was censored. However, all of the cells in our health data were from >10 patients, and thus the data were free from the anonymization process.

In addition, as such restrictions existed, we could not handle any relatively small population as an independent group. For example, children are known to be susceptible to pollutants and should be handled independently [22], but the number of children in the DPC survey data was too small to pass the anonymization process.

We used the following conditions for the aggregation of the individual data. For patients, we included patients admitted due to a non-accidental cause (ICD-10: A00 to R99) by their primary diagnosis in order to exclude patients admitted for an injury. We included patients regardless of their DPC codes and included those who were not in the DPC codes. We excluded patients who were transferred from another hospital or who were on a planned admission. We also excluded patients who were admitted to hospitals outside their city of residence in order to reduce exposure misclassification. Appendix A is the patient selection flow chart. For hospitals, we included hospitals that participated in the DPC survey in 2015, 2016, and 2017. We included the year 2017 since the data of late March 2016 were collected in the DPC survey in 2017. We excluded hospitals that only partially participated during these 3 years. A final total of 628 hospitals was included.

From the DPC survey data, we obtained the number of hospital admissions on day t ($HA_{t}$) and the corresponding total charges billed during admission on day t ($TC_{t}$) for each city. Using these data, we calculated the daily mean total charges per patient on day t ($TCPP_{t}$) by the following formula and used the results in the subsequent analysis.
(1)$$TCPP_{t}=TC_{t}/HA_{t}$$

### 2.3. Hospital Data

As the DPC survey data do not include all of the hospitals in each city, the data’s exact representativeness is unclear. We defined the cover ratios as the percentage of the hospitals, the hospital beds, and the patients included in this study to each city’s total and estimated them by using data from another national survey. From 2018, the Japanese government obligated hospitals with hospital beds to submit a yearly predefined report called the Hospital Bed Function Report (HBFR). The report consists of statistics concerning each hospital’s roles and functions, including the number of hospital beds and the number of newly admitted patients in the previous year.

We obtained data from the hospitals’ HBFRs in 2018 from the MHLW. The data’s target period was from 1 July 2015 to 30 June 2016, which was in the midst of our study period. We estimated cover ratios of the following from the data: (1) hospitals, (2) DPC hospitals and non-DPC hospitals, (3) hospital beds, and (4) newly admitted patients.

There are several points to consider regarding the data from the HBFRs. First, the patients’ selection criteria in the HBFR differ from our health data and included more patients. For example, the HBFRs include injury patients, but we excluded them. Second, a few hospitals did not submit any HBFRs to the MHLW. Of the 628 included hospitals, eight hospitals’ data were not present in the HBFR data.

### 2.4. Environmental Data

We obtained the hourly ambient air measurements of PM_2.5_, suspended particulate matter (SPM), photochemical oxidants (O_x_), nitrogen dioxide (NO_2_), and sulfur dioxide (SO_2_) for each of the 12 cities from Japan’s National Institute for Environmental Studies. All of the cities had more than one ambient air monitoring station that measured all of the included air pollutants. These monitoring stations were placed at sites that represented the ambient exposure of the community [14,23]. For each monitoring station, we calculated daily mean concentrations of each pollutant by averaging the hourly measurements if >18 hourly measurements were available. We used the mean of all available results as a proxy for the exposure value in each city. Since there have been no routine measurements of ambient PM_10_ in Japan, we defined coarse particulate matter (coarse PM) as the difference between SPM and PM_2.5_ [14].

We also obtained daily mean ambient temperature and relative humidity data from the Japan Meteorological Agency. One city (Saitama) had no data for temperature, and two cities (Saitama and Kawasaki) had no data for relative humidity. For these cities, we used the data measured at the nearest city in the same prefecture.

### 2.5. Statistical Analysis

We used a two-stage approach to determine the associations between ambient PM_2.5_ and (1) hospital admissions, (2) the total charges, and (3) the daily mean total charges per patient. In the first stage, we built city-specific models for each outcome of interest. We used generalized additive models with quasi-Poisson regression for the hospital admissions and a log-linear regression for the daily mean of total charges per patient. In the second stage, we used a random-effects meta-analysis with the restricted maximum likelihood estimation method to obtain the national average estimates from the first stage. We used Cochran’s Q test and the *I*^2^ statistics to examine heterogeneity between cities [24].

We included the following confounders: (1) a natural cubic smooth function of calendar time with seven degrees of freedom (*df*) per year for long-term trends, including infectious disease epidemics [10,25]; (2) a natural spline function of the 3 day moving average temperature (6 *df*) and relative humidity (3 *df*) to adjust for weather conditions [25,26]; and (3) indicator variables for the days of the week and public holidays.

For the exposure measurements, we used the single-day lags of the same day (lag 0) and previous days (lag 1 and 2) and the 2 day moving average of the current and previous days (lag 01) [14,26]. For the subsequent analyses, we used the lag with the largest t-statistics [27]. We also used distributed lag models (DLMs) to examine the temporal associations [28]. We specified the lagged effect of PM_2.5_ with a second-degree polynomial function [29] with an extension of the lag period for up to 6 days. We also used unconstrained DLMs, which give noisy but unbiased estimates [30]. We followed previous work to obtain the national average for DLMs [31].

The final model for hospital admission is below:(2)$$\ln\left[E\left(HA_{t}\right)\right]=\alpha +\beta s\left(pm_{t},\ldots ,pm_{t-L}\right)+ns\left(time,\;df=7/year\times 2years\right)\;+DOW_{t}+holiday_{t}+ns\left(temp_{t},df=6\right)+ns\left(rh_{t},df=3\right)$$
where $E\left(HA_{t}\right)$ is the expected number of hospital admissions on day t; α is the intercept; β are parameters of interest; $s\left(.\right)$ is a function that specifies the lag-response associations with a maximum lag up to L days before; $pm_{t}$ is the PM_2.5_ concentration on day t; $ns\left(.\right)$ is the natural cubic spline function; time is the calendar time; $DOW_{t}$ is a categorical variable for the day of the week; $holiday_{t}$ is an indicator variable of public holidays on day t; $temp_{t}$ is the 3 day moving average temperature on day t; and $rh_{t}$ is the mean relative humidity on day t. For the cost analysis, we replaced the left-hand side of the equation with $E\left[\ln\left(TC_{t}\right)\right]$ or $E\left[\ln\left(TCPP_{t}\right)\right]$, which represent the expected log-transformed total charges or expected log-transformed daily mean total charges per patient on day t, respectively.

We performed stratification analyses by age (<65 or ≥65 years) and sex. We also divided the cities into those in East Japan and West Japan at the longitude of 138°. We used z-tests for the differences between groups [32]. However, we considered a difference by a factor of ≥2 as important regardless of the statistical significance [33].

We conducted the following sensitivity analyses: (1) two-pollutant models with additional adjustment for coarse PM, O_x_, NO_2_, and SO_2_, and (2) the use of alternative *df* values. We present the estimated effects and 95% confidence interval (95% CI) as the percent change (PC) per 10 µg/m^3^ increase in the PM_2.5_ concentration. A *p*-value < 0.05 was considered significant. We performed all analyses using R (ver. 3.6.3) [34]. We used the “metafor” package to conduct the meta-analysis [24]. In the DLM analysis, we used the “dlnm” package [35] and “mvmeta” package [31].

## 3. Results

### 3.1. Descriptive Analysis

A total of 2,017,750 hospital admissions in the 12 Japanese cities during the period from 1 April 2015 to 31 March 2017 were included in this study. Table 1 and Table 2 summarize the characteristics of the 12 cities. The daily mean number of admissions ranged from 70.9 to 834.3. The mean of the daily ambient concentrations of PM_2.5_ ranged from 7.9 µg/m^3^ to 15.8 µg/m^3^.

Appendix A shows the city-specific estimates of cover ratios by data from the HBFRs, and Table 3 shows the cover ratio in the entire included cities. The estimated cover ratios were as follows: DPC hospitals, 99.2%; non-DPC hospitals, 22.8%; emergency hospital beds, 79.6%; and emergency care (EC) patients, 88.0%.

### 3.2. Regression Analysis

Appendix A provide the city-specific estimates of the associations between the ambient PM_2.5_ concentrations and the hospital admissions, the total charges, and the daily mean total charges per patient, and Table 4 provides the nationally averaged results. We observed low-to-moderate between-city heterogeneity in all of the outcomes, and all were insignificant. The largest t-statistic was observed at lag 01 for hospital admissions and total charges and lag 2 for the daily mean total charges per patient. We estimated an increase of 0.56% (95% CI: 0.14–0.99%) for hospital admissions and an increase of 1.17% (95% CI: 0.44–1.90%) for total charges at lag 01. We estimated an increase of 0.75% (95% CI: 0.34–1.16%) for the daily mean total charges per patient at lag 2.

Figure 1 presents the national average estimates of the associations by DLM constrained with a second-degree polynomial. We observed lagged effects of 1–3 days for the daily mean total charges per patient. Appendix A presents the results of the estimation by the unconstrained DLM. We observed similar temporal associations but with wide confidence intervals.

### 3.3. Stratification and Sensitivity Analysis

Table 5 indicates the estimated effects modified by age, sex, and the location of the cities. We observed stronger associations with hospital admissions among males (0.74%, 95% CI: 0.18–1.30%) than females (0.36%, 95% CI: −0.13–0.85%) and in East Japan (0.97%, 95% CI: 0.10–1.84%) than West Japan (0.29%, 95% CI: −0.27–0.86%). We observed stronger associations with total charges among patients aged ≥65 years (1.53%, 95% CI: 0.74–2.32%) than patients aged <65 years (0.04%, 95% CI: −1.05–1.13%), and the difference was statistically significant. We observed stronger associations with daily mean total charges per patient among patients aged ≥65 years (0.90%, 95% CI: 0.43–1.37%) than patients aged <65 years (0.27%, 95% CI: −0.36–0.91%) and in East Japan (1.21%, 95% CI: 0.51–1.92%) than in West Japan (0.40%, 95% CI: −0.11–0.92%).

Appendix A shows the estimates from the two-pollutant models. We observed estimates that were similar to those in the single pollutant models. However, the association with total charges was not significant after controlling for SO_2_. In Appendix A, we provide the estimates when alternative *df* values were used; similar estimates were obtained after changing the *df* values of calendar time, temperature, and humidity.

## 4. Discussion

In this time-series study, we examined the associations between ambient PM_2.5_ concentrations and hospital admissions, total charges, and daily mean total charges per patient in 12 cities in Japan. We observed significant positive associations in all of the outcomes.

We estimated that a 10 µg/m^3^ increase in the ambient PM_2.5_ concentration was associated with a 0.56% (95% CI: 0.14–0.99%) increase in all-cause hospital admissions. These results are generally consistent with findings in previous multi-city studies and meta-analyses, although those data were limited to admissions due to cardiovascular and respiratory diseases. A study of 213 counties in the U.S. estimated a 0.65% (95% CI: 0.48–0.83%) increase in cardiovascular admissions and a 0.25% (95% CI: 0.01–0.48%) increase in respiratory admissions due to ambient PM_2.5_ exposure [4]. A study of eight European cities reported a 0.51% (95% CI: 0.12–0.90%) increase in cardiovascular admissions and a 1.36% (95% CI: 0.23 to 2.49%) increase in respiratory admissions due to ambient PM_2.5_ exposure [5]. In addition, a meta-analysis of studies of Chinese populations estimated a 0.37% (95% CI: 0.17–0.56%) increase in cardiovascular admissions and a 0.51% (95% CI: 0.23–0.79%) increase in respiratory admissions due to ambient PM_10_ exposure [7].

Few studies have examined the relationship between ambient PM_2.5_ exposure and all-cause hospital admissions [6,10], and the majority of the studies were based on single cities [36,37,38,39,40] and are thus susceptible to publication bias [41]. The few existing multi-city studies of the association between ambient PM_2.5_ exposure and all-cause hospital admissions obtained estimated risks that tended to be similar to our present findings. A recent study of 24 Canadian cities estimated a 0.29% (95% CI: 0.03–0.56%) increase in all-cause hospital admissions due to ambient PM_2.5_ exposure [42]. A recent study of 200 Chinese cities estimated a 0.20% (95% CI: 0.08–0.31%) increase due to ambient PM_2.5_ exposure [10].

The association between ambient PM_2.5_ concentrations and hospital admission costs in an entire hospitalized population has not been established. We observed a significant positive association between the ambient PM_2.5_ concentration and total charges. In accord with studies of specific disease groups [12,13], we also observed a significant positive association between PM_2.5_ concentrations and the total charges per patient. Our results imply that ambient PM_2.5_ exposure before a hospital admission event was associated with the subsequent total charge of the admission. However, earlier investigations on costs often used fixed per-patient costs and implicitly did not consider the variation [8,9,43]. As their results neglected an increase in per-patient costs, these studies might have underestimated the associations between ambient PM_2.5_ concentrations and hospital admission costs.

We observed lagged effects for the association between ambient PM_2.5_ concentrations and the daily mean total charges per patient. Previous studies have reported a latency of several days for respiratory diseases, possibly due to systemic inflammation and/or immune suppression [44,45]. The onset of respiratory diseases after admission or a decline of the patient’s general condition by these mechanisms may have caused the observed lagged effect. Further studies with individual-level data are needed to clarify this issue.

We also observed spatial heterogeneity in the association of ambient PM_2.5_ concentrations; for both hospital admissions and the daily mean total charges per patient, the magnitude of the estimates in West Japan was more than twice that of East Japan. These results are consistent with those of a study of mortality linked to ambient PM_2.5_ exposure in 100 cities in Japan [14]. This tendency may be partly attributable to the variation in chemical compositions of ambient PM_2.5_ [46]. However, the scarcity of data on chemical compositions hampered further investigation.

There are several study limitations to note. First, our estimates from HBFR data showed that our health data included 99.2% of the DPC hospitals, where most acute inpatients were treated [17], and 88.0% of emergency care patients. These estimates justify the use of our health data in the current study. However, we included only 22.8% of the non-DPC hospitals in Japan. As such, our sample might not have fairly represented the entire hospitalized population in Japan. Second, we did not consider most of the effects of individual-level factors, since only anonymized aggregated data were available. Third, we only considered part of the medical costs. We did not consider costs after hospital discharge or indirect market costs. Not accounting for these costs might have biased our estimates downwards. Fourth, we did not specifically examine the patients’ clinical courses after their hospital admissions. We could not determine which components of medical costs were associated with ambient PM_2.5_ exposure. In addition, ignoring in-hospital deaths might have affected our estimates downwards since in-hospital deaths can reduce total charges by artificially shortening the length of stay in hospitals [47]. Fifth, we used the average of all monitoring stations in each city as a proxy for personal ambient exposure. This exposure assessment may have caused an exposure measurement error and biased our estimates downwards [48]. Lastly, this study included only populated urban cities, and thus the generalizability of the results is limited [49].

## 5. Conclusions

We observed significant associations between ambient PM_2.5_ concentrations and all-cause hospital admissions, the corresponding total charges, and the total charges per patient.

## Figures and Tables

**Figure 1 ijerph-18-04116-f001:**
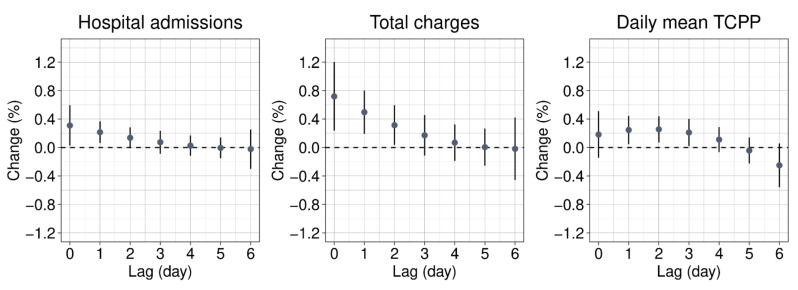
National average percent change of all-cause hospital admissions, total charges, and daily mean total charges per patient with a 10 μg/m^3^ increase in the PM_2.5_ concentration estimated by the distributed lag model constrained with a second-degree polynomial. TCPP: total charges per patient.

**Table 1 ijerph-18-04116-t001:** Descriptive statistics on the population, total all-cause hospital admissions, number of included hospitals, and number of air pollution monitoring stations in 12 cities in Japan.

Region	City	Population, Millions	No. of Hospitals	No. of Hospital Admissions Per Day ^a^		No. of Stations
				Mean	SD	
West Japan	Sapporo	2.0	77	210.8	61.4	3
	Sendai	1.1	24	104.2	27.3	2
	Saitama	1.3	13	70.9	19.5	2
	Tokyo’s 23 wards	9.3	160	834.3	180.6	10
	Yokohama	3.7	53	303.3	66.0	17
	Kawasaki	1.5	18	105.5	24.8	8
East Japan	Nagoya	2.3	42	235.2	58.0	4
	Kyoto	1.5	37	163.5	36.5	3
	Osaka	2.7	72	314.3	75.6	6
	Kobe	1.5	49	163.1	43.8	2
	Hiroshima	1.2	31	103.7	28.6	4
	Fukuoka	1.5	52	151.4	38.3	2
-	Combined	29.5	628	2760.3	208.7	63

^a^ Admitted patients who were not transferred or planned admissions for a non-accidental cause (ICD-10: A00 to R99) from 1 April 2015 to 31 March 2017 from the DPC survey data specifically extracted for our study. SD: standard deviation; ICD-10: the 10th edition of the International Classification of Disease; DPC: Diagnosis Procedure Combination.

**Table 2 ijerph-18-04116-t002:** Descriptive statistics on daily mean environmental factors in 12 cities in Japan.

City	PM_2.5_, μg/m^3^		Coarse PM, μg/m^3^		O_x_, ppb		NO_2_, ppb		SO_2_, ppb		Temperature, °C		Relative Humidity, %	
	Mean	SD	Mean	SD	Mean	SD	Mean	SD	Mean	SD	Mean	SD	Mean	SD
Sapporo	7.9	5.2	4.1	2.5	28.5	10.0	10.0	7.4	0.9	0.8	9.4	9.5	66.5	11.5
Sendai	11.3	6.4	2.5	3.4	30.8	10.5	10.3	5.2	0.5	0.4	13.6	8.2	68.0	13.7
Saitama	11.7	6.5	8.6	4.2	28.6	11.9	14.3	6.8	1.8	0.6	16.0	8.2	62.3	16.6
Tokyo’s 23 wards	13.7	7.4	4.9	4.7	26.8	11.6	18.7	8.1	2.0	1.0	16.4	7.7	67.9	16.6
Yokohama	12.4	6.4	8.5	4.9	27.6	12.2	16.0	7.3	2.3	0.8	16.8	7.4	69.0	16.0
Kawasaki	12.9	7.0	3.7	4.8	28.9	12.1	18.0	7.9	1.5	0.9	16.8	7.4	69.0	16.0
Nagoya	13.3	6.9	6.5	5.2	30.8	12.8	15.5	6.8	1.2	0.7	16.7	8.2	65.3	13.4
Kyoto	11.9	6.4	3.1	3.1	30.4	11.5	10.4	4.8	3.0	1.0	16.8	8.4	66.2	10.5
Osaka	15.5	7.4	5.4	3.7	27.0	11.7	19.9	8.1	3.8	1.8	17.5	8.0	65.4	11.2
Kobe	13.0	6.7	4.9	5.9	29.4	11.4	15.9	7.2	2.1	1.4	17.5	7.8	64.6	11.1
Hiroshima	13.8	7.0	9.2	6.2	28.4	11.6	11.3	4.6	1.5	0.8	16.9	8.0	64.8	11.2
Fukuoka	15.8	7.4	6.5	4.5	30.8	11.6	14.6	5.7	2.1	1.4	17.7	7.5	72.4	12.1
Combined	12.8	7.0	5.7	5.0	29.0	11.7	14.6	7.5	1.9	1.4	16.0	8.3	66.8	13.8

PM: particulate matter, O_x_: photochemical oxidants, NO_2_: nitrogen dioxide, SO_2_: sulfur dioxide.

**Table 3 ijerph-18-04116-t003:** Cover ratios estimated by data from hospital bed function report (HBFR) in 2018.

Variable	Total	Included	%
DPC hospitals ^a^	374	371	99.2
Non-DPC hospitals	1094	249	22.8
Hospital beds	252,118	168,327	66.8
Emergency hospital beds	199,996	159,182	79.6
Patients ^b^	4,731,059	4,110,501	86.9
EC patients ^c^	1,642,797	1,445,426	88.0

^a^ Hospitals that were under the DPC/PDPS. ^b^ Newly admitted patients from 1 July 2015 to 30 June 2016. ^c^ Newly admitted EC patients who were not transferred or planned admissions from 1 July 2015 to 30 June 2016. DPC: diagnosis procedure combination, DPC/PDPS: diagnostic procedure combination/per-diem payment system, EC: emergency care.

**Table 4 ijerph-18-04116-t004:** National average percent change of all-cause hospital admissions, total charges, and daily mean total charges per patient with a 10 μg/m^3^ increase in the ambient PM_2.5_ concentration estimated by the single-day or moving average lag model.

Lag	Hospital Admissions				Total Charges				Daily Mean TCPP			
	PC ^a^	95% CI	*I* ^2^	*p*	PC ^a^	95% CI	*I* ^2^	*p*	PC ^a^	95% CI	*I* ^2^	*p*
Lag 0	0.41	0.05–0.76	1.8	0.026	0.82	0.29–1.35	0.0	0.002	0.25	−0.20–0.70	28.1	0.272
Lag 1	0.41	0.05–0.77	1.2	0.026	0.86	0.23–1.48	22.3	0.007	0.43	−0.08–0.94	42.8	0.101
Lag 2	0.16	−0.20–0.51	0.0	0.385	0.68	0.15–1.22	0.0	0.011	0.75	0.34–1.16	13.9	<0.001
Lag 3	0.31	−0.04–0.66	0.5	0.087	0.34	−0.19–0.87	1.6	0.207	0.20	−0.16–0.55	0.0	0.283
Lag 01	0.56	0.14–0.99	4.1	0.010	1.17	0.44–1.90	23.1	0.002	0.44	−0.10–0.97	30.1	0.108

^a^ Adjusted for calendar time, temperature, relative humidity, public holidays, and the day of the week. TCPP: total charges per patient, PC: percent change.

**Table 5 ijerph-18-04116-t005:** National average percent change of all-cause hospital admissions, total charges, and daily mean total charges per patient with a 10 μg/m^3^ increase in the ambient PM_2.5_ concentration stratified by age, sex, and city of residence (lag 01 for hospital admission and total charges, and lag 2 for the daily mean total charges per patient).

Subgroups		Hospital Admissions			Total Charges			Daily Mean TCPP		
		PC ^a^	95% CI	*p* ^b^	PC ^a^	95% CI	*p* ^b^	PC ^a^	95% CI	*p* ^b^
Age				0.620			0.030			0.124
	<65 yrs	0.45	−0.11–1.01		0.04	−1.05–1.13		0.27	−0.36–0.91	
	≥65 yrs	0.63	0.16–1.11		1.53	0.74–2.32		0.90	0.43–1.37	
Sex				0.314			0.884			0.692
	Male	0.74	0.18–1.30		1.18	0.34–2.04		0.66	0.02–1.29	
	Female	0.36	−0.13–0.85		1.09	0.17–2.02		0.83	0.25–1.42	
Region				0.205			0.424			0.071
	West Japan	0.29	−0.27–0.86		0.93	0.09–1.78		0.40	−0.11–0.92	
	East Japan	0.97	0.10–1.84		1.61	0.19–3.06		1.21	0.51–1.92	

^a^ Adjusted for calendar time, temperature, relative humidity, public holidays, and the day of the week. ^b^ The *p*-values were obtained from z-tests between the subgroups. TCPP: total charges per patient, PC: percent change.

## Data Availability

The dataset analyzed during the current study is not publicly available due to data protection requirements from Japan’s MHLW.

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
