# Peer review of "Short-Term Associations of Ambient Fine Particulate Matter (PM2.5) with All-Cause Hospital Admissions and Total Charges in 12 Japanese Cities"

_ijerph, 2021, doi:10.3390/ijerph18084116_

Round 1

Reviewer 1 Report

This study performed a time-series design with generalized additive models to evaluate the short-term associations between fine particulate matter (PM2.5) and all-cause hospital admissions and total charges in 12 Japanese cities, where are located throughout the length of Japan. To some extent, this study provides new evidence regarding the economic burden of PM2.5 for entire hospitalized populations in Japan. Besides, author observed significant positive associations at levels below the current Japanese air quality standard, and took this as one of the main conclusions.

Major concerns:

  1. The author used generalizedadditive models (GAM) with quasi-Poisson regression for hospital admissions and generalized additive models (GAM) with log-linear regression for total charges, but the author did not give exact mathematical expression for this tow models. This makes it difficult for readers to understand how the author's results were obtained.
  2. In section "Health data", the description of 86-96 lines, it is suggested that the author use a flow chart to visualize the screening process of patient data.
  3. In section "Hospitaldata", the author estimated the hospital information by using data from HBFR, but unable to clarify its contribution to this study.
  1. Since theauthor said "We observed associations at levels below the current Japanese air quality standard " ,did the author draw the main conclusion from table 5? If so, the author should give the calculation details of the trend tests used in the article. In addition, the RRs and 95%CIs of two categorized PM2.5 concentrations(25–35 µg/m3 and >35 µg/m3 ) for hospital admissions were too close, but the trend tests were significant. Please reexamine the calculation .

Minor concerns:

  1. "we used a random-effects meta-analysis to obtain the national average estimates from the firststage. We report I2 statistics for heterogeneity " , for this statement, it would be better to specify this analysis process, such as: Fine particulate matter constituents and cause-specific mortality in China: A nationwide modelling study.Environ.Int.2020,143,0160-4120,doi:10.1016/j/envint.2020.105927.

2. Line 130: Statical analysis?

Reviewer 2 Report

1. Many pollutants were mentioned in the Methods section (section 2.4), but then they are not shown in the results or conclusions section. Could you clarify this point.
2. Mention if the pollutant monitors are in rural, urban, or both areas. How do the authors know that the patients were exposed to the ambient concentrations measured by the monitors and that they are part of the hospital admissions database (HBFR )?
3. Indoor concentrations are different from ambient concentrations. The study does not distinguish between outdoor and indoor. Clarify this point in the manuscript and mainly in the introduction and methods.
4. Separate the results section into subsections, similar to the methods section.

Reviewer 3 Report

The aim of the paper was to assess the short-term association, up to lag 3, between exposure to PM2.5 and overall hospital admissions in a sample of 628 hospitals located in 12 Japanese cities, with a population that exceeds 1 million inhabitants. The assessed outcomes were both the daily frequency of hospital admissions and the daily average per-capita associated charge.

I think that the paper is definitely worthy of consideration for two reasons: it assesses the impact of air pollution in relatively low-concentration setting, and it analyzes costs, a dimension that has been rarely investigated so far, and usually through impact assessment rather than directly modelling costs. It is therefore quite novel in the field. I found the paper well written and clear, and authors should be commended for that.

However, I have a few major points I would suggest to consider before publication:

  1. Lines 98-99 – Authors state “As individual-level data were not available, we used the daily mean total charges per patient as a dependent variable”. I would ask the authors to clarify if they had the cumulative charges and they divided them by the number of admission, or if they had the aggregated data in term of average per-capita charge. In case they had the cumulative charges, I would ask them to clarify why they decided to use the per-patient charge as a dependent variable: these two choices underlie two slightly different concepts. Indeed, using the strategy of modeling the cumulative charges leads to assessing the overall increase in the expenditures, that is directly relatable to the increase observed in the total number of admissions. On the other hand, assessing the increase in the per-patient expenditure reveals if the hospitalization per-se was more expensive when PM concentration was higher. Such increase refers to the single hospitalization, and it should not be compared with the increase in the frequency of hospitalization (Lines ). I would suggest to review results and discussion in light of this consideration.
  2. Lines 143-146, Table 4 – Since it has been demonstrated that the delay in PM effect depends on the outcome (e.g. cardiovascular effects are immediate, while respiratory effects might occur 2-3 days after the exposure), I would suggest the authors to consider using a distributed lag model, in order to highlight the shape of the delayed effect of the pollutants.
  3. Lines 146-149, Table 5, lines 255-260 – I think that the analysis stratified by PM2.5 concentration classes is hampered by the limited number of observations for the classes 25-35 and >= 35. In table 1, authors report that such classes include 7% and 5.4% of all observations, respectively. Results reported in table 5 are therefore necessarily affected by low power: this would justify the inconsistency of RR with the p-value for trend. I would suggest to either consider removing the analysis, or to use only the WHO standard as a cut-off.

I also have some minor notes:

  1. Line 29 – The reference [1] is a little old, I would suggest to add a more recent reference, such as “Thurston GD et al. A joint ERS/ATS policy statement: what constitutes an adverse health effect of air pollution? An analytical framework. Eur Respir J. 2017 Jan 11;49(1):1600419. doi: 10.1183/13993003.00419-2016.”
  2. Line 101 – I would suggest to clarify the concept of cover ratio.
  3. Line 117 – Please, rephrase “suspected particulate matter” as “suspended particulate matter”.
  4. Lines 132-133 – I would suggest to rephrase as “In the first stage, we built city-specific models for each outcome of interest”, or another similar statement.
  5. Lines 240-246 - Even though these considerations about the cause underlying the differences with the estimates form other studies are valid, they refer to a difference that does not appear statistically significant.
  6. Lines 253-254. Could the authors please clarify why they state they might have underestimated the associations with costs?
  7. Line 277 - Could the authors please clarify why they state that ignoring in-hospital deaths might have affected our estimates downwards?

Round 2

Reviewer 1 Report

According to the comments, the author has carefully revised the deficiencies. The overall description of the study and the discussion of the results were also relatively comprehensive. 

Reviewer 3 Report

Authors should be commended for thoroughly revising their paper.

They have addressed all the points I raised in a very satisfactory way.

I think that the paper has consistently improved and that authors did a very good job.